# How the First Year of the COVID-19 Pandemic Impacted Patients’ Hospital Admission and Care in the Vascular Surgery Divisions of the Southern Regions of the Italian Peninsula

**DOI:** 10.3390/jpm12071170

**Published:** 2022-07-19

**Authors:** Eugenio Martelli, Giovanni Sotgiu, Laura Saderi, Massimo Federici, Giuseppe Sangiorgi, Matilde Zamboni, Allegra R. Martelli, Giancarlo Accarino, Giuseppe Bianco, Francesco Bonanno, Umberto M. Bracale, Enrico Cappello, Giovanni Cioffi, Giovanni Colacchio, Adolfo Crinisio, Salvatore De Vivo, Carlo Patrizio Dionisi, Loris Flora, Giovanni Impedovo, Francesco Intrieri, Luca Iorio, Gabriele Maritati, Piero Modugno, Mario Monaco, Giuseppe Natalicchio, Vincenzo Palazzo, Fernando Petrosino, Francesco Pompeo, Raffaele Pulli, Davide Razzano, Maurizio R. Ruggieri, Carlo Ruotolo, Paolo Sangiuolo, Gennaro Vigliotti, Pietro Volpe, Antonella Biello, Pietro Boggia, Michelangelo Boschetti, Enrico M. Centritto, Flavia Condò, Lucia Cucciolillo, Amodio S. D’Amodio, Mario De Laurentis, Claudio Desantis, Daniela Di Lella, Giovanni Di Nardo, Angelo Disabato, Ilaria Ficarelli, Angelo Gasparre, Antonio N. Giordano, Alessandro Luongo, Mafalda Massara, Vincenzo Molinari, Andrea Padricelli, Marco Panagrosso, Anna Petrone, Serena Pisanello, Roberto Prunella, Michele Tedesco, Alberto M. Settembrini

**Affiliations:** 1Department of General and Specialist Surgery, Sapienza University of Rome, 155 Viale del Policlinico, 00161 Rome, Italy; 2Saint Camillus International University of Health Sciences, 8 Via di Sant’Alessandro, 00131 Rome, Italy; 3Division of Vascular Surgery, S. Anna and S. Sebastiano Hospital, Via F. Palasciano, 81100 Caserta, Italy; apadricelli@inwind.it; 4Clinical Epidemiology and Medical Statistics Unit, Department of Medicine, Surgery and Pharmacy, University of Sassari, Viale San Pietro, 07100 Sassari, Italy; gsotgiu@uniss.it (G.S.); lsaderi@uniss.it (L.S.); 5Department of Systems Medicine, University of Rome Tor Vergata, 1 Viale Montpellier, 00133 Rome, Italy; federicm@uniroma2.it; 6Department of Biomedicine and Prevention, University of Rome Tor Vergata, 1 Viale Montpellier, 00133 Rome, Italy; gsangiorgi@gmail.com; 7Division of Vascular Surgery, Saint Martin Hospital, 22 Viale Europa, 32100 Belluno, Italy; 8Medicine and Surgery School of Medicine, Campus Bio-Medico University of Rome, 21 Via À. del Portillo, 00128 Rome, Italy; allegramartelli02@gmail.com; 9San Giovanni Di Dio e Ruggi d’Aragona Hospital, Via San Leonardo s.n.c., 84125 Salerno, Italy; gcaccarino@tin.it; 10San Giovanni Bosco Hospital, 225 Via F. M. Briganti, 80144 Naples, Italy; peppebianco20@libero.it (G.B.); michelangelo.boschetti@gmail.com (M.B.); 11Madonna delle Grazie Hospital, Via Montescaglioso s.n.c., 75100 Matera, Italy; bonannofr@virgilio.it; 12Federico II Polyclinic, Department of Public Health and Residency Program in Vascular Surgery, University of Naples Federico II, 5 Via S. Pansini, 80131 Naples, Italy; umbertomarcello.bracale@unina.it (U.M.B.); flaviacondo2@gmail.com (F.C.); marcopanagrosso@gmail.com (M.P.); annapetrone91@gmail.com (A.P.); 13Mediterranean Neurological Institute NEUROMED, 18 Via Atinense, 80122 Naples, Italy; enrico.cappello@yahoo.it (E.C.); pompeo@neuromed.it (F.P.); 14Pellegrini Hospital, 41 Via Portamedina alla Pignasecca, 80134 Naples, Italy; g.cioffi99@virgilio.it (G.C.); sdevivo71@gmail.com (S.D.V.); asdamodio@alice.it (A.S.D.); 15F. Miulli Hospital, Km. 4100 Strada Provinciale 127 Acquaviva-Santeramo, 70021 Acquaviva delle Fonti, Italy; gm.colacchio@gmail.com (G.C.); anggas@libero.it (A.G.); 16Salus Clinic, 4 Via F. Confalonieri, 84091 Battipaglia, Italy; adolfocrinisio71@gmail.com (A.C.); dinardogiovanni47@gmail.com (G.D.N.); 17Cardinal Panico Hospital, 4 Via San Pio X, 73039 Tricase, Italy; cpdionisi@libero.it (C.P.D.); angelodisabato@gmail.com (A.D.); 18San Giuseppe Moscati Hospital, Contrada Amoretta, 83100 Avellino, Italy; loflora60@gmail.com; 19SS. Annunziata Hospital, 1 Via F. Bruno, 74121 Taranto, Italy; impedovovasc@libero.it (G.I.); serenapisanello@libero.it (S.P.); robertoprunella@icloud.com (R.P.); 20Annunziata Hospital, 1 Via Migliori, 87100 Cosenza, Italy; f.intrieri@tiscali.it (F.I.); vincenzomolinari79@gmail.com (V.M.); 21Cardarelli Hospital, 1 Via U. Petrella, 86100 Campobasso, Italy; luca.iorio@asrem.org (L.I.); lucia.cucciolillo@asrem.org (L.C.); 22A. Perrino Hospital, Strada Statale 7 per Mesagne, 72100 Brindisi, Italy; gabrielemaritati@hotmail.com (G.M.); pietroboggiafr@gmail.com (P.B.); 23Gemelli Molise Hospital, Catholic University of the Sacred Heart, 1 Largo A. Gemelli, 86100 Campobasso, Italy; piero.modugno@gmail.com (P.M.); emcentritto@yahoo.it (E.M.C.); 24Pineta Grande Hospital, Km. 30 Via Domitiana, 81030 Castelvolturno, Italy; mariomonaco55@libero.it; 25Venere Hospital, 1 Via Ospedale di Venere, 70131 Bari, Italy; pino.natalicchio@gmail.com (G.N.); bielloantonella@gmail.com (A.B.); 26Casa Sollievo della Sofferenza Hospital, Viale Cappuccini s.n.c., 71013 San Giovanni Rotondo, Italy; palaenzo@katamail.com (V.P.); antonionicolagiordano@gmail.com (A.N.G.); 27San Luca Hospital, 1 Via F. Cammarota, 84078 Vallo della Lucania, Italy; fernandopetrosino01@gmail.com (F.P.); alessandroluongo_md@libero.it (A.L.); 28Polyclinic of Bari, Department of Emergency and Organs Transplantation, Aldo Moro University of Bari, 11 Piazza Giulio Cesare, 70124 Bari, Italy; raffaele.pulli@uniba.it (R.P.); claudio.desantis25@gmail.com (C.D.); 29San Pio Hospital, 1 Via dell’angelo, 82100 Benevento, Italy; razzanodavide@libero.it; 30Riuniti Polyclinic, 1 Viale L. Pinto, 71122 Foggia, Italy; mauruggieri@gmail.com (M.R.R.); mtedesco@ospedaliriunitifoggia.it (M.T.); 31Cardarelli Hospital, 9 Via A. Cardarelli, 80131 Naples, Italy; carlo.ruotolo@aocardarelli.it (C.R.); iladott@libero.it (I.F.); 32Monaldi Hospital, Via L. Bianchi s.n.c., 84100 Naples, Italy; paolo.sangiuolo@ospedalideicolli.it (P.S.); mario.delaurentis@yahoo.it (M.D.L.); 33Del Mare Hospital, 11 Via E. Russo, 80147 Naples, Italy; gevigli@tin.it (G.V.); danieladilella@libero.it (D.D.L.); 34Bianchi-Melacrino-Morelli Hospital, 21 Via G. Melacrino, 89124 Reggio di Calabria, Italy; pietro.volpe257@gmail.com (P.V.); drmafaldamassara@gmail.com (M.M.); 35Division of Vascular Surgery, Maggiore Polyclinic Hospital Ca’ Granda IRCCS and Foundation, 28 Via F. Sforza, 20122 Milan, Italy; amsettembrini@gmail.com

**Keywords:** COVID-19, carotid stenosis, abdominal aortic aneurysm, chronic limb-threatening ischemia, amputation, deep venous thrombosis

## Abstract

Background: To investigate the effects of the COVID-19 lockdowns on the vasculopathic population. Methods: The Divisions of Vascular Surgery of the southern Italian peninsula joined this multicenter retrospective study. Each received a 13-point questionnaire investigating the hospitalization rate of vascular patients in the first 11 months of the COVID-19 pandemic and in the preceding 11 months. Results: 27 out of 29 Centers were enrolled. April-December 2020 (7092 patients) vs. 2019 (9161 patients): post-EVAR surveillance, hospitalization for Rutherford category 3 peripheral arterial disease, and asymptomatic carotid stenosis revascularization significantly decreased (1484 (16.2%) vs. 1014 (14.3%), *p* = 0.0009; 1401 (15.29%) vs. 959 (13.52%), *p* = 0.0006; and 1558 (17.01%) vs. 934 (13.17%), *p* < 0.0001, respectively), while admissions for revascularization or major amputations for chronic limb-threatening ischemia and urgent revascularization for symptomatic carotid stenosis significantly increased (1204 (16.98%) vs. 1245 (13.59%), *p* < 0.0001; 355 (5.01%) vs. 358 (3.91%), *p* = 0.0007; and 153 (2.16%) vs. 140 (1.53%), *p* = 0.0009, respectively). Conclusions: The suspension of elective procedures during the COVID-19 pandemic caused a significant reduction in post-EVAR surveillance, and in the hospitalization of asymptomatic carotid stenosis revascularization and Rutherford 3 peripheral arterial disease. Consequentially, we observed a significant increase in admissions for urgent revascularization for symptomatic carotid stenosis, as well as for revascularization or major amputations for chronic limb-threatening ischemia.

## 1. Introduction

The Coronavirus Disease 2019 (COVID-19) pandemic has changed lifestyles and working activities worldwide. Following the publication of the Italian government decree in March 2020, three-month strict lockdown measures were implemented countrywide to avoid social contact. Hospital-related routines were interrupted to prioritize the management of COVID-19 cases; in particular, outpatient and elective surgeries were postponed.

Similar prevention and public health interventions were implemented from mid-September to the beginning of December 2020 in response to the second wave of the pandemic. Furthermore, except for situations of proven urgency, the quality of diagnostic and therapeutic care in general medicine in Southern Italy was impacted negatively during the lockdowns, affecting the diagnosis, management, and surveillance of vascular patients. For instance, screening programs for carotid stenosis and abdominal aortic aneurysm and early detection of Rutherford category 3 peripheral arterial disease (R3-PAD) worsening towards chronic limb-threatening ischemia (CLTI) have surely been dramatically postponed. 

This study was conducted to assess the eventual impact of suspension of hospitalization for elective vascular surgery on the incidence rates of hospital admissions for complications caused by common vascular conditions compared to the pre-pandemic period. 

## 2. Materials and Methods

A multicenter retrospective study was conducted through a cross-sectional survey; the majority of public vascular surgery wards and those accredited with the National Health System (NHS) located in the south of the Italian peninsula were enrolled, i.e., the regions of Campania, Molise, Basilicata, Puglia, and Calabria (population: 12,646,486; area: 62,809 km^2^, Figure 1). 

Even though healthcare policies are issued at the regional level in Italy, the above-mentioned regions implemented similar COVID-19 restrictions.

Twenty-seven vascular surgery divisions joined the study; only two centers (one public and one private) declined to participate due to lack of human resources for data collection. A 13-item questionnaire was provided, asking about the number of patients that underwent: (1)open repair or endovascular aneurysm repair (EVAR) for asymptomatic abdominal aortic aneurysm (AAA);(2)open repair or EVAR for primary ruptured or symptomatic AAA;(3)duplex or computed-tomography scans performed for post-EVAR surveillance;(4)Previous EVAR treated again (in an open or endovascular fashion) for recurring symptomatic or ruptured AAA, or for endoleak at risk of AAA rupture (type 1, 3, or 2 with sac expansion);(5)open, or endovascular treatments for thrombotic, non-embolic, acute lower limb ischemia;(6)treatments for Rutherford category 3 peripheral arterial disease (R3-PAD) in socially active patients with very short distance intermittent claudication (less than 50 mt. on the flat), not responsive to best medical therapy, and asking for a resolutive treatment to improve their lifestyle;(7)open or endovascular revascularizations for chronic limb-threatening ischemia (CLTI);(8)CLTI patients who have had a thigh or leg amputated;(9)open or endovascular revascularizations for asymptomatic severe internal carotid artery (ICA) stenosis;(10)Asymptomatic severe ICA stenosis on surgical waiting list, complicated to total obstruction (with or without neurological symptoms);(11)symptomatic ICA stenosis operated in urgency;(12)conservative or surgical treatments for venous ulcers;(13)diagnosis of deep vein thrombosis (DVT), also from requests of consultation from the emergency room or any medical/surgical divisions.

The aim was to compare these vascular surgery activities before (i.e., 11 months pre-COVID-19) and during (i.e., 11 months from the beginning of the pandemic) the COVID-19 pandemic.

Indications for carotid, AAA, CLTI, and venous surgery, as well as the diagnosis of vascular diseases, are those reported in the current, well-known, international guidelines. 

Informed consent for the present study was waived because of the retrospective and aggregated nature of the study analysis. Being an observational study, according to Italian law mandatory approval is not needed. 

Formal ethical approval and patient informed consent were not needed. The current Italian legislation on observational studies such as the present one does not request the above-mentioned documents when clinical data are anonymized (Official Gazette of the Italian Republic # 76, 31 March 2008). 

Clinical characteristics were described with absolute and relative (percentage) frequencies. Qualitative variables were compared using the chi-square test. Percentage differences for the collected variables (delta) between the pre-COVID-19 and COVID-19 periods were computed. A two-tailed *p*-value < 0.05 was considered statistically significant. All statistical analyses were carried out using STATA software version 17 (StataCorp LLC, College Station, TX, USA).

## 3. Results

Information on 19,603 cases was collected: 11,129 (56.8%) during the pre-COVID-19 period and 8474 (43.2%) during the COVID-19 period (Table 1).

Imaging for post-EVAR surveillance, frequency of admissions for R3-PAD, and asymptomatic ICA stenosis revascularization significantly decreased (16.2% vs. 14.3%, *p* = 0.0009; 15.29% vs. 13.52%, *p* = 0.0006; 17.01% vs. 13.17%, *p* < 0.0001, respectively) during the COVID-19 period (from April to December 2020) compared to the same time-period of the previous pre-pandemic year. During the COVID-19 period, admissions for open repair or EVAR for primary ruptured or symptomatic AAA, open or endovascular revascularization for CLTI, major amputations for CLTI, urgent revascularization for symptomatic ICA stenosis, and diagnosis of DVT significantly increased (2.41% vs. 1.91%, *p* = 0.03; 16.98% vs. 13.59%, *p* < 0.0001; 5.01% vs. 3.91%, *p* = 0.0007; 2.16% vs. 1.53%, *p* = 0.0009; 9.8% vs. 8.22%, *p* = 0.0004, respectively, Table 2 and Figure 2).

When only April 2019 and April 2020 were compared, the significant decrease in imaging for post-EVAR surveillance, frequency of admissions for R3-PAD, and asymptomatic ICA stenosis revascularization was confirmed (19.43%, vs. 9.98% *p* < 0.0001; 18.06% vs. 9.38%, *p* < 0.0001; 17.11% vs. 10.38%, *p*: 0.0006, respectively), as well as the significant increase in admissions for open or endovascular revascularization for CLTI, major amputations for CLTI, and diagnosis of DVT (17.76% vs. 4.86%, *p* < 0.0001; 8.58% vs. 4.65%, *p* = 0.002; 13.37% vs. 9.19%, *p* = 0.01, respectively). Furthermore, a significant decrease in admissions of patients requiring further treatment after EVAR (1.69% vs. 1%, *p* = 0.006) and a significant increase in the admissions of patients treated for acute, thrombotic lower limb ischemia was found (8.78% vs. 4.96%, *p* = 0.005, Table 3).

The comparison between May–June 2019 and May–June 2020 highlighted similar decreases in imaging for post-EVAR surveillance and frequency of admissions for asymptomatic ICA stenosis revascularization (17.04% vs. 12.16%, *p* = 0.0001, and 16.77% vs. 14.14%, *p* = 0.03, respectively) and increases in admissions for open repair or EVAR for primary ruptured or symptomatic AAA, urgent revascularization for symptomatic ICA stenosis, and diagnosis of DVT (2.84% vs. 1.5%, *p* = 0006; 1.98% vs. 1.05%, *p* = 0.01; 10.71% vs. 7.84%, *p* = 0.002, respectively, Table 4).

July–August 2019 vs. July–August 2020 showed the same significant decrease in the frequency of admissions for asymptomatic carotid stenosis revascularization (16.96% vs. 12.79%, *p* = 0.001) as well as the same significant increase in the admissions for urgent revascularization for symptomatic ICA stenosis (2.42% vs. 1.38%, *p* = 0.04, Table 5). 

There was a constant significant decrease in the frequency of admissions for asymptomatic ICA stenosis revascularization (17.07% vs. 13.72%, *p* = 0.002) in September–October 2019 vs. September–October 2020 (Table 6). 

The comparison of November-December 2019 and November-December 2020 showed the same significant decrease in the frequency of admissions for R3-PAD and asymptomatic ICA stenosis revascularization (15.4% vs. 12.98%, *p* = 0.04, and 17.18% vs. 12.8%, *p* < 0.0002, respectively), as well as the same significant increase in admissions for open or endovascular revascularization for CLTI, major amputations for CLTI, and urgent revascularization for symptomatic ICA stenosis (17.29% vs. 14.14%, *p* = 0.007; 4.49% vs. 3.09%, *p* = 0.02; 2.52% vs. 1.69%, *p* = 0.09, respectively, Table 7). 

Groupings of the initial months of 2020 (January–February vs. March–April) were characterized by a significant decrease in imaging for post-EVAR surveillance, frequency of admissions for R3-PAD, and asymptomatic ICA stenosis revascularization (16.11% vs. 10.89%, *p* = 0.0001; 15.4% vs. 11.73%, *p* = 0.005; 16.36% vs. 10.99%, *p* = 0.0001, respectively), while an increase of admissions for major amputations for CLTI and diagnoses of DVT was found (6.89% vs. 3.61%, *p* < 0.0001 and 12.1% vs. 8.79%, *p* = 0.004, respectively). Furthermore, a significant increase in treatment of acute thrombotic lower limb ischemia occurred (7.91% vs. 5.49%, *p* = 0.01, Table 8).

When comparing only January 2020 and January 2021, the frequency of admissions for R3-PAD and asymptomatic ICA stenosis revascularization significantly decreased (16.75% vs. 11.37%, *p* = 0.001, and 15.69% vs. 10.75%, *p* = 0.002, respectively), and admissions for open or endovascular revascularization for CLTI and major amputations for CLTI significantly increased (17.55% vs. 13.49%, *p* = 0.02, and 5.44% vs. 3.25%, *p* = 0.02, respectively, Table 9). 

During the pandemic period from March 2020 to January 2021, 36/1713 (2.1%) patients who presented with CLTI, 33/501 (6.6%) patients who required major amputation, and 9/207 (4.4%) patients with stroke or transient ischemic attacks tested positive for COVID-19.

Treatment for venous ulcers and frequency of admissions for EVAR or open repair for asymptomatic AAA and severe ICA stenosis >80% (according to the European Carotid Surgery Trial, ECST, parameters) on operating waiting lists complicated to total obstruction (detected at duplex control before revascularization or because they became symptomatic) did not change throughout the study period.

## 4. Discussion

Early identification of life-threatening vascular conditions, which are mostly asymptomatic, is essential. Changes during the COVID-19 pandemic could have affected patient prognosis. Several researchers, clinicians, and policymakers have been trying to understand the real impact of the pandemic on clinical activities [1].

In the Netherlands it has been reported that during the lockdown period of 16 March until 30 April 2020, there was a statistically significant increase in CLTI severity and rates of major amputations compared to the same time period during the two previous years. No difference in vascular surgical care for patients with an AAA has been observed [2]. On the contrary, a study carried out in the metropolitan city of Bologna, Italy, focusing on the first 30 days of the COVID-19 pandemic showed that the number of surgical interventions was similar to that recorded in 2018 and 2019. No differences were found in the acute/emergency setting, including interventions for acute ischemia, although SARS-CoV-2 infections triggers thrombogenic mechanisms [3]. At the same time, English colleagues have reported different results [4]. These two last conflicting experiences are probably affected by the limited period of time and/or the population analyzed.

A US cross-sectional study focusing on the period from 14–24 April 2020 showed a significant impact on the practice of vascular surgery across the country, with an unprecedented number of surgical cases cancelled and changes in on-call schedules. The majority of continued elective cases were on aortic repair and maintenance of dialysis function rather than peripheral arterial disease or venous procedures [5]. Similarly, in Indochina almost all vascular interventions were suspended during the COVID-19 outbreak [6].

Our multicenter study covering more than one-fifth of the Italian geographical area and population over a longer time-period (11 months before and 11 months during the COVID-19 pandemic) showed a significant decrease in elective interventions for the following:−Prophylactic ICA revascularization during each month of the pandemic compared to the prior year, as well as during the first two months of the pandemic compared to the prior two months;−Imaging for post-EVAR surveillance from April to June, 2020 compared to the corresponding time-period in 2019, as well as during the first two months of the pandemic (March–April, 2020) in comparison with the two months before it (January–February, 2020);−Treatment for R3-PAD during the first two months of the pandemic in comparison with the two prior months, and in April 2020 and January 2021 when compared with the corresponding month of the previous year.

On the other hand, there was a significant increase in diagnosis of DVT and frequency of admission for urgent revascularization for symptomatic ICA stenosis, revascularization for CLTI, major amputations, open repair or EVAR for primary ruptured or symptomatic AAA, and treatment of acute thrombotic lower limb ischemia.

The decrease in admissions for prophylactic ICA revascularization could be associated with the increased hospitalization rate for urgent revascularization of symptomatic carotid stenosis.

Furthermore, the decrease in admissions for R3-PAD in the first two months of the pandemic could explain the increased rate of hospitalization for revascularization and major amputation of CLTI patients in April and November-December 2020 and January 2021. Around 20% of patients with intermittent claudication experience deterioration of limb status over a five-year period, and symptomatic deterioration is greatest within the first year after diagnosis [7]. 

Interestingly, in January 2021, when the immediate pandemic restrictions were lifted, a major decrease in admissions for R3-PAD and severe asymptomatic ICA stenosis persisted compared to pre-pandemic levels. Our analysis suggests that these delays may have further consequences in the coming months. Project 1 (Impact of COVID-19 on scheduled vascular operations) of the international Vascular Surgery COVID-19 Collaborative (VASCC) registry aims to answer this particular question. The VASCC is a combined international effort to obtain prospective data on the impact of widespread vascular surgical care delays due to an international crisis or pandemic [8,9]. An increased rate of DVT during the first four months of the pandemic and of hospitalization for thrombotic acute lower limb ischemia recalls the prothrombotic effects of the SARS-CoV-2 infection [10,11]. This broad spectrum of clinical manifestations, affecting almost all organs and systems, is a consequence of endothelial dysfunction and systemic inflammatory response. Endothelial cells activated by a hyperinflammatory state induced by viral infection may promote localized inflammation, increase reactive oxidative species production, and alter the dynamic interplay between procoagulant and fibrinolytic factors in the vascular system, leading to thrombotic disease both in the pulmonary circulation and in peripheral veins and arteries [12].

Although the US national trends in Vascular Surgical Practice showed a decreased rate of urgent and emergency aortic and carotid interventions, our study described an increased rate of open repair or EVAR for ruptures or symptomatic AAA and of symptomatic carotid stenosis treated with urgency [13].

The constant trends of patients who underwent EVAR or open repair for primary asymptomatic AAA during the current pandemic could be associated with the positive organization of healthcare delivery in the participating centers, although no specific data were collected to support this hypothesis. Similar explanations could support the trends of conservative or surgical treatment for venous ulcers, although they are managed in wards other than vascular surgery (i.e., vascular medicine and dermatology).

Several study limitations can be highlighted: several vascular diseases (e.g., thoracic or thoraco-abdominal aortic aneurysms and dialysis access) were not considered. Complex aortic procedures are often referred to specialist centers, and we thought that the numbers would be too low. In Italy, arteriovenous fistulas are performed by nephrologists; likewise, varicose vein surgery was excluded based on its postponement caused by low priority. We evaluated only the first eleven months of the COVID-19 pandemic against the corresponding 2019 months; as such, inter-annual variability cannot be excluded. Stratification of the findings based on SARS-CoV-2 positivity was not always performed; infection could have increased the incidence of certain vascular diseases (e.g., DVT). Asymptomatic severe ICA stenosis that progressed to occlusion (and thus was managed non-surgically) could have been missed, as it can cause cerebral ischemia, which can be managed in different medical wards (e.g., stroke unit, intensive care unit, neurology, internal medicine) and thus be under-reported.

## 5. Conclusions

The interruption of elective surgery during the COVID-19 pandemic caused decreased rates of post-EVAR surveillance and hospitalization for prophylactic carotid revascularization and R3-PAD. These findings are associated with an increased rate of hospital admission for urgent revascularization for symptomatic carotid stenosis, CLTI, and subsequent major amputations.

The vascular community is called upon to raise awareness of the dangers arising from restrictions in the management of these elective vascular patients during the pandemic crisis. 

The long-term effects on the management of vascular patients should be evaluated in the near future.

## Figures and Tables

**Figure 1 jpm-12-01170-f001:**
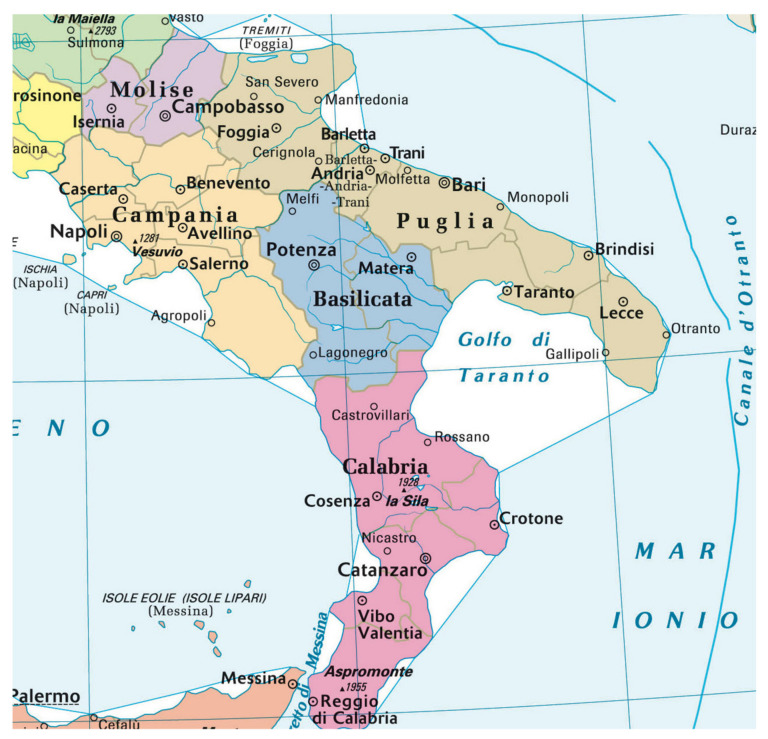
The five regions of the southern Italian peninsula (reproduced with permission from Atlante Geografico Mondiale, Milan, Italy: Touring Club Italiano, 2021).

**Figure 2 jpm-12-01170-f002:**
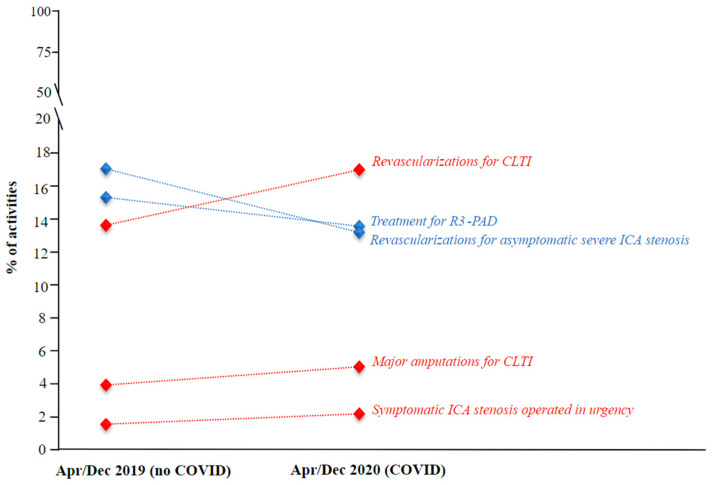
Graphic of the main results for the COVID-19 period from April to December 2020 compared with the non-COVID-19 period from April to December 2019. CLTI, chronic limb-threatening is-chemia; R3-PAD, Rutherford category 3 peripheral arterial disease; ICA, internal carotid artery. Blue indicates elective procedures; red indicates urgent procedures.

**Table 1 jpm-12-01170-t001:** Summary of the responses to the questionnaire.

a. The pre-COVID period.											
Time Period→	April 2019	May 2019	June 2019	July 2019	August 2019	September 2019	October 2019	November 2019	December 2019	January 2020	February 2020
Questions ↓											
#1	99	108	93	94	45	102	125	120	105	102	90
#2	184	182	192	159	77	173	197	175	145	162	155
#3	16	12	14	10	6	16	11	10	11	11	10
#4	171	180	152	153	88	162	166	179	150	175	128
#5	47	45	53	51	40	53	53	58	55	59	49
#6	57	71	67	68	53	58	85	75	54	67	52
#7	87	90	82	77	76	79	97	86	79	96	77
#8	162	180	188	183	99	189	190	207	160	164	158
#9	16	21	12	12	15	19	18	30	32	18	17
#10	46	169	165	145	108	151	159	166	136	141	136
#11	44	48	45	41	40	36	38	32	34	34	37
#12	0	1	2	0	0	2	1	0	1	0	1
#13	18	9	14	15	8	20	20	22	14	16	13
**b. The COVID-19 period.**											
**Time Period→**	**March** **2020**	**April** **2020**	**May** **2020**	**June** **2020**	**July 2020**	**August** **2020**	**September 2020**	**October** **2020**	**November** **2020**	**December** **2020**	**January 2021**
**Questions** **↓**											
#1	69	53	58	73	81	52	97	109	87	81	85
#2	67	50	69	115	131	76	149	149	142	133	144
#3	5	5	8	11	10	8	14	9	8	7	6
#4	79	47	78	125	118	80	153	147	113	98	92
#5	41	44	37	45	37	42	47	42	38	38	44
#6	32	25	47	61	68	41	67	48	61	43	61
#7	63	67	71	91	86	71	86	85	68	70	76
#8	66	52	90	124	124	66	126	144	115	93	87
#9	14	14	22	21	18	16	24	23	15	18	15
#10	90	89	121	138	152	92	160	171	142	139	142
#11	31	43	36	41	44	36	44	38	42	31	44
#12	4	0	0	1	0	0	1	1	2	0	0
#13	12	12	19	11	21	15	16	18	23	18	13

**Table 2 jpm-12-01170-t002:** Summary of the responses to the questionnaire for the COVID-19 period from April to December 2020 compared with the non-COVID-19 period from April to December 2019.

	Activities	April/ December 2019 (no COVID) (n = 9161) n (%)	April/ December 2020 (COVID) (n = 7092) n (%)	*p*-Value	Delta %
#1	Open repair/EVAR for asymptomatic AAA	891 (9.73)	691 (9.74)	1.00	−22.45
#2	Open repair/EVAR for primary ruptured or symptomatic AAA	175 (1.91)	171 (2.41)	0.03	−2.29
#3	Post-EVAR surveillance	1484 (16.2)	1014 (14.3)	0.0009	−31.67
#4	Previous EVAR treated again for recurring symptomatic or ruptured AAA, or endoleak type 1, 3, or 2 with sac expansion	106 (1.16)	80 (1.13)	0.55	−24.53
#5	Treatment for thrombotic acute lower limb ischemia	455 (4.97)	370 (5.22)	0.57	−18.68
#6	Treatment for R3-PAD	1401 (15.29)	959 (13.52)	0.0006	−31.55
#7	Revascularizations for CLTI	1245 (13.59)	1204 (16.98)	<0.0001	−3.29
#8	Major amputations for CLTI	358 (3.91)	355 (5.01)	0.0007	−0.84
#9	Revascularizations for asymptomatic severe ICA stenosis	1558 (17.01)	934 (13.17)	<0.0001	−40.05
#10	Asymptomatic severe ICA stenosis on surgical waiting list complicated to total obstruction	7 (0.08)	5 (0.07)	0.91	−28.57
#11	Symptomatic ICA stenosis operated in urgency	140 (1.53)	153 (2.16)	0.0009	9.29
#12	Treatment for venous ulcers	588 (6.42)	461 (6.5)	0.80	−21.60
#13	Diagnosis of deep vein thrombosis	753 (8.22)	695 (9.8)	0.0004	−7.70

EVAR, endovascular aneurysm repair; AAA, abdominal aortic aneurysm; R3-PAD, Rutherford category 3 peripheral arterial disease; CLTI, chronic limb-threatening ischemia; ICA, internal carotid artery.

**Table 3 jpm-12-01170-t003:** Summary of the responses to the questionnaire for the COVID-19 month of April 2020 compared with the non-COVID-19 month of April 2019.

	Activities	April 2019 (no COVID) (n = 947) n (%)	April 2020 (COVID) (n = 501) n (%)	*p*-Value	Delta %
#1	Open repair/EVAR for asymptomatic AAA	99 (10.45)	53 (10.58)	0.91	−46.46
#2	Open repair/EVAR for primary ruptured or symptomatic AAA	16 (1.69)	14 (2.79)	0.16	−12.50
#3	Post-EVAR surveillance	184 (19.43)	50 (9.98)	<0.0001	−72.83
#4	Previous EVAR treated again for recurring symptomatic or ruptured AAA, or endoleak type 1, 3, or 2 with sac expansion	16 (1.69)	5 (1)	0.006	−68.75
#5	Treatment for thrombotic acute lower limb ischemia	47 (4.96)	44 (8.78)	0.005	−6.38
#6	Treatment for R3-PAD	171 (18.06)	47 (9.38)	<0.0001	−72.51
#7	Revascularizations for CLTI	46 (4.86)	89 (17.76)	<0.0001	93.48
#8	Major amputations for CLTI	44 (4.65)	43 (8.58)	0.002	−2.27
#9	Revascularizations for asymptomatic severe ICA stenosis	162 (17.11)	52 (10.38)	0.0006	−67.90
#10	Asymptomatic severe ICA stenosis on surgical waiting list complicated to total obstruction	0 (0)	0 (0)	-	-
#11	Symptomatic ICA stenosis operated in urgency	18 (1.9)	12 (2.4)	0.52	−33.33
#12	Treatment for venous ulcers	57 (6.02)	25 (4.99)	0.43	−56.14
#13	Diagnosis of deep vein thrombosis	87 (9.19)	67 (13.37)	0.01	−22.99

EVAR, endovascular aneurysm repair; AAA, abdominal aortic aneurysm; R3-PAD, Rutherford category 3 peripheral arterial disease; CLTI, chronic limb-threatening ischemia; ICA, internal carotid artery.

**Table 4 jpm-12-01170-t004:** Summary of the responses to the questionnaire for the COVID-19 months of May-June 2020 compared with the non-COVID-19 months of May-June 2019.

	Activities	May–June 2019 (no COVID) (n = 2195) n (%)	May–June 2020 (COVID) (n = 1513) n (%)	*p*-Value	Delta %
#1	Open repair/EVAR for asymptomatic AAA	201 (9.16)	131 (8.66)	0.60	−34.83
#2	Open repair/EVAR for primary ruptured or symptomatic AAA	33 (1.5)	43 (2.84)	0.006	30.30
#3	Post-EVAR surveillance	374 (17.04)	184 (12.16)	0.0001	−50.80
#4	Previous EVAR treated again for recurring symptomatic or ruptured AAA, or endoleak type 1, 3, or 2 with sac expansion	26 (1.18)	19 (1.26)	0.79	−26.92
#5	Treatment for thrombotic acute lower limb ischemia	98 (4.46)	82 (5.42)	0.21	−16.33
#6	Treatment for R3-PAD	332 (15.13)	203 (13.42)	0.15	−38.86
#7	Revascularizations for CLTI	334 (15.22)	259 (17.12)	0.12	−22.46
#8	Major amputations for CLTI	93 (4.24)	77 (5.09)	0.20	−17.20
#9	Revascularizations for asymptomatic severe ICA stenosis	368 (16.77)	214 (14.14)	0.03	−41.85
#10	Asymptomatic severe ICA stenosis on surgical waiting list complicated to total obstruction	3 (0.14)	1 (0.07)	1.00	−66.67
#11	Symptomatic ICA stenosis operated in urgency	23 (1.05)	30 (1.98)	0.01	30.43
#12	Treatment for venous ulcers	138 (6.29)	108 (7.14)	0.34	−21.74
#13	Diagnosis of deep vein thrombosis	172 (7.84)	162 (10.71)	0.002	−5.81

EVAR, endovascular aneurysm repair; AAA, abdominal aortic aneurysm; R3-PAD, Rutherford category 3 peripheral arterial disease; CLTI, chronic limb-threatening ischemia; ICA, internal carotid artery.

**Table 5 jpm-12-01170-t005:** Summary of the responses to the questionnaire for the COVID-19 months of July–August 2020 compared with the non-COVID-19 months of July–August 2019.

	Activities	July–August 2019 (no COVID) (n = 1663) n (%)	July–August 2020 (COVID) (n = 1485) n (%)	*p*-Value	Delta %
#1	Open repair/EVAR for asymptomatic AAA	139 (8.36)	133 (8.96)	0.55	−4.32
#2	Open repair/EVAR for primary ruptured or symptomatic AAA	27 (1.62)	34 (2.29)	0.15	25.93
#3	Post-EVAR surveillance	236 (14.19)	207 (13.94)	0.81	−12.29
#4	Previous EVAR treated again for recurring symptomatic or ruptured AAA, or endoleak type 1, 3, or 2 with sac expansion	16 (0.96)	18 (1.21)	0.59	12.50
#5	Treatment for thrombotic acute lower limb ischemia	91 (5.47)	79 (5.32)	0.80	−13.19
#6	Treatment for R3-PAD	241 (14.49)	198 (13.33)	0.33	−17.84
#7	Revascularizations for CLTI	253 (15.21)	244 (16.43)	0.36	−3.56
#8	Major amputations for CLTI	81 (4.87)	80 (5.39)	0.53	−1.23
#9	Revascularizations for asymptomatic severe ICA stenosis	282 (16.96)	190 (12.79)	0.001	−32.62
#10	Asymptomatic severe ICA stenosis on surgical waiting list complicated to total obstruction	0 (0)	0 (0)	-	-
#11	Symptomatic ICA stenosis operated in urgency	23 (1.38)	36 (2.42)	0.04	56.52
#12	Treatment for venous ulcers	121 (7.28)	109 (7.34)	1.00	−9.92
#13	Diagnosis of deep vein thrombosis	153 (9.2)	157 (10.57)	0.19	2.61

EVAR, endovascular aneurysm repair; AAA, abdominal aortic aneurysm; R3-PAD, Rutherford category 3 peripheral arterial disease; CLTI, chronic limb-threatening ischemia; ICA, internal carotid artery.

**Table 6 jpm-12-01170-t006:** Summary of the responses to the questionnaire for the COVID-19 months of September-October 2020 compared with the non-COVID-19 months of September-October 2019.

	Activities	September/ October 2019 (no COVID) (n = 2220) n (%)	September/ October 2020 (COVID) (n = 1968) n (%)	*p*-Value	Delta %
#1	Open repair/EVAR for asymptomatic AAA	227 (10.23)	206 (10.47)	0.75	−9.25
#2	Open repair/EVAR for primary ruptured or symptomatic AAA	37 (1.67)	47 (2.39)	0.11	27.03
#3	Post-EVAR surveillance	370 (16.67)	298 (15.14)	0.16	−19.46
#4	Previous EVAR treated again for recurring symptomatic or ruptured AAA, or endoleak type 1, 3, or 2 with sac expansion	27 (1.22)	23 (1.17)	1.00	−14.81
#5	Treatment for thrombotic acute lower limb ischemia	106 (4.77)	89 (4.52)	0.65	−16.04
#6	Treatment for R3-PAD	328 (14.77)	300 (15.24)	0.72	−8.54
#7	Revascularizations for CLTI	310 (13.96)	331 (16.82)	0.01	6.77
#8	Major amputations for CLTI	74 (3.33)	82 (4.17)	0.12	10.81
#9	Revascularizations for asymptomatic severe ICA stenosis	379 (17.07)	270 (13.72)	0.002	−28.76
#10	Asymptomatic severe ICA stenosis on surgical waiting list complicated to total obstruction	3 (0.14)	2 (0.1)	1.00	−33.33
#11	Symptomatic ICA stenosis operated in urgency	40 (1.8)	34 (1.73)	0.81	−15.00
#12	Treatment for venous ulcers	143 (6.44)	115 (5.84)	0.42	−19.58
#13	Diagnosis of deep vein thrombosis	176 (7.93)	171 (8.69)	0.35	−2.84

EVAR, endovascular aneurysm repair; AAA, abdominal aortic aneurysm; R3-PAD, Rutherford category 3 peripheral arterial disease; CLTI, chronic limb-threatening ischemia; ICA, internal carotid artery.

**Table 7 jpm-12-01170-t007:** Summary of the responses to the questionnaire for the COVID-19 months of November/December 2020 compared with the non-COVID-19 months of November/December 2019.

	Activities	November/ December 2019 (no COVID) (n = 2136) n (%)	November/ December 2020 (COVID) (n = 1625) n (%)	*p*-Value	Delta %
#1	Open repair/EVAR for asymptomatic AAA	225 (10.53)	168 (10.34)	0.84	−25.33
#2	Open repair/EVAR for primary ruptured or symptomatic AAA	62 (2.9)	33 (2.03)	0.08	−46.77
#3	Post-EVAR surveillance	320 (14.98)	275 (16.92)	0.11	−14.06
#4	Previous EVAR treated again for recurring symptomatic or ruptured AAA, or endoleak type 1, 3, or 2 with sac expansion	21 (0.98)	15 (0.92)	0.75	−28.57
#5	Treatment for thrombotic acute lower limb ischemia	113 (5.29)	76 (4.68)	0.4	−32.74
#6	Treatment for R3-PAD	329 (15.4)	211 (12.98)	0.04	−35.87
#7	Revascularizations for CLTI	302 (14.14)	281 (17.29)	0.007	−6.95
#8	Major amputations for CLTI	66 (3.09)	73 (4.49)	0.02	10.61
#9	Revascularizations for asymptomatic severe ICA stenosis	367 (17.18)	208 (12.8)	0.0002	−43.32
#10	Asymptomatic severe ICA stenosis on surgical waiting list complicated to total obstruction	1 (0.05)	2 (0.12)	0.14	100.00
#11	Symptomatic ICA stenosis operated in urgency	36 (1.69)	41 (2.52)	0.09	13.89
#12	Treatment for venous ulcers	129 (6.04)	104 (6.4)	0.61	−19.38
#13	Diagnosis of deep vein thrombosis	165 (7.72)	138 (8.49)	0.37	−16.36

EVAR, endovascular aneurysm repair; AAA, abdominal aortic aneurysm; R3-PAD, Rutherford category 3 peripheral arterial disease; CLTI, chronic limb-threatening ischemia; ICA, internal carotid artery.

**Table 8 jpm-12-01170-t008:** Summary of the responses to the questionnaire for the COVID-19 months of March/April 2020 compared with the non-COVID-19 months of January/February 2020.

	Activities	January/ February 2020 (no COVID) (n = 1968) n (%)	March/ April 2020 (COVID) (n = 1074) n (%)	*p*-Value	Delta %
#1	Open repair/EVAR for asymptomatic AAA	192 (9.76)	122 (11.36)	0.17	−36.46
#2	Open repair/EVAR for primary ruptured or symptomatic AAA	35 (1.78)	28 (2.61)	0.14	−20.00
#3	Post-EVAR surveillance	317 (16.11)	117 (10.89)	0.0001	−63.09
#4	Previous EVAR treated again for recurring symptomatic or ruptured AAA, or endoleak type 1, 3, or 2 with sac expansion	21 (1.07)	10 (0.93)	0.60	−52.38
#5	Treatment for thrombotic acute lower limb ischemia	108 (5.49)	85 (7.91)	0.01	−21.30
#6	Treatment for R3-PAD	303 (15.4)	126 (11.73)	0.005	−58.42
#7	Revascularizations for CLTI	277 (14.08)	179 (16.67)	0.06	−35.38
#8	Major amputations for CLTI	71 (3.61)	74 (6.89)	<0.0001	4.23
#9	Revascularizations for asymptomatic severe ICA stenosis	322 (16.36)	118 (10.99)	0.0001	−63.35
#10	Asymptomatic severe ICA stenosis on surgical waiting list complicated to total obstruction	1 (0.05)	4 (0.37)	0.08	300.00
#11	Symptomatic ICA stenosis operated in urgency	29 (1.47)	24 (2.23)	0.16	−17.24
#12	Treatment for venous ulcers	119 (6.05)	57 (5.31)	0.43	−52.10
#13	Diagnosis of deep vein thrombosis	173 (8.79)	130 (12.1)	0.004	−24.86

EVAR, endovascular aneurysm repair; AAA, abdominal aortic aneurysm; R3-PAD, Rutherford category 3 peripheral arterial disease; CLTI, chronic limb-threatening ischemia; ICA, internal carotid artery.

**Table 9 jpm-12-01170-t009:** Summary of the responses to the questionnaire for the COVID-19 month of January 2021 compared with the non-COVID-19 month of January 2020.

	Activities	January 2020 (no COVID) (n = 1045) n (%)	January 2021 (COVID) (n = 809) n (%)	*p*-Value	Delta %
#1	Open repair/EVAR for asymptomatic AAA	102 (9.76)	85 (10.51)	0.62	−16.67
#2	Open repair/EVAR for primary ruptured or symptomatic AAA	18 (1.72)	15 (1.85)	0.87	−16.67
#3	Post-EVAR surveillance	162 (15.5)	144 (17.8)	0.19	−11.11
#4	Previous EVAR treated again for recurring symptomatic or ruptured AAA, or endoleak type 1, 3, or 2 with sac expansion	11 (1.05)	6 (0.74)	0.49	−45.45
#5	Treatment for thrombotic acute lower limb ischemia	59 (5.65)	44 (5.44)	0.85	−25.42
#6	Treatment for R3-PAD	175 (16.75)	92 (11.37)	0.001	−47.43
#7	Revascularizations for CLTI	141 (13.49)	142 (17.55)	0.02	0.71
#8	Major amputations for CLTI	34 (3.25)	44 (5.44)	0.02	29.41
#9	Revascularizations for asymptomatic severe ICA stenosis	164 (15.69)	87 (10.75)	0.002	−46.95
#10	Asymptomatic severe ICA stenosis on surgical waiting list complicated to total obstruction	0 (0)	0 (0)	-	
#11	Symptomatic ICA stenosis operated in urgency	16 (1.53)	13 (1.61)	0.86	−18.75
#12	Treatment for venous ulcers	67 (6.41)	61 (7.54)	0.35	−8.96
#13	Diagnosis of deep vein thrombosis	96 (9.19)	76 (9.39)	0.88	−20.83

EVAR, endovascular aneurysm repair; AAA, abdominal aortic aneurysm; R3-PAD, Rutherford category 3 peripheral arterial disease; CLTI, chronic limb-threatening ischemia; ICA, internal carotid artery.

## Data Availability

Raw data were obtained from the vascular surgery divisions of the southern regions of the Italian peninsula and are readily available for presentation to the referees and the editors of the journal, if requested.

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
