# Peer review of "How the First Year of the COVID-19 Pandemic Impacted Patients’ Hospital Admission and Care in the Vascular Surgery Divisions of the Southern Regions of the Italian Peninsula"

_jpm, 2022, doi:10.3390/jpm12071170_

Round 1

Reviewer 1 Report

In the conclusion section of the abstract section, elective activities are to be changed to elective procedures, or surgeries.

In the visual abstract section, the representative color of the graphs should be defined. Whether before the pandemic is represented with red should be clarified.

Page5, the second half of to be revised to beyond mid-September

Page 5, Furthermore, except for situations of proven urgency, the quality of diagnostic and therapeutic care of General Medicine in Southern Italy was impacted negatively during the lockdowns, affecting the diagnosis, management, and surveillance of vascular patients.

Notwithstanding the fact that COVID affects patients with vascular disease, the evidence should explicitly be explained. The sentence starts with general medicine and lasts with vasculopathy, which eliminated strong interconnections. 

Page 7, Informed consent for the present study was waived because of the retrospective and aggregated nature of the study analysis.

The retrospective design of the study is not an appropriate explanation for informed consent waiver. Institutional Review Board or ethics committee reviews are mandatory in all retrospective studies.

The main concept that should be addressed is not only the frequency of categorized management but also the frequency of admission that seems to be definitively affected by the Pandemic. 

Early identification of life-threatening vascular conditions, which are mostly asymptomatic, is key, to be revised to is critical or essential. 

Author Response

Extensive editing of English language and style has been performed.

Two more references have been added.

In the conclusion section of the abstract section, elective activities are to be changed to elective procedures, or surgeries.

R: The change has been done.

In the visual abstract section, the representative color of the graphs should be defined. Whether before the pandemic is represented with red should be clarified.

R: “Blue indicates elective procedures; red indicates urgent procedures” has been added in the legend of the Graphical abstract.

Page5, the second half of to be revised to beyond mid-September

R: The change has been done.

Page 5, Furthermore, except for situations of proven urgency, the quality of diagnostic and therapeutic care of General Medicine in Southern Italy was impacted negatively during the lockdowns, affecting the diagnosis, management, and surveillance of vascular patients.

Notwithstanding the fact that COVID affects patients with vascular disease, the evidence should explicitly be explained. The sentence starts with general medicine and lasts with vasculopathy, which eliminated strong interconnections.

R: The following paragraph has been added.For instance, screening programs for carotid stenosis and abdominal aortic aneurysm, or early detection of Rutherford category 3 peripheral arterial disease (R3-PAD) worsening towards chronic limb-threatening ischemia (CLTI) surely have been dramatically posticipated.”.

Page 7, Informed consent for the present study was waived because of the retrospective and aggregated nature of the study analysis.

The retrospective design of the study is not an appropriate explanation for informed consent waiver. Institutional Review Board or ethics committee reviews are mandatory in all retrospective studies.

R: “Informed consent for the present study was waived because of the retrospective and aggregated nature of the study analysis. Being an observational study, according to the Italian law, a mandatory approval is not needed” has been changed in “Formal ethical approval and patient informed consent were not needed. The current Italian legislation on observational studies (like the present one) does not request the above-mentioned documents when clinical data are anonymized (Official Gazette of the Italian Republic # 76, March 31, 2008)”.

The main concept that should be addressed is not only the frequency of categorized management but also the frequency of admission that seems to be definitively affected by the Pandemic. 

R: The title of the manuscript has been changed accordingly.

In the Background of the Abstract section was already reported “Each received a 13-point questionnaire, investigating the hospitalization rate of vascular patients…”.

“…the admissions for…” was added in the Results and Conclusions of the Abstract section, where needed.

The last paragraph of the Introduction section changed: “This study was conducted to assess the eventual impact of suspension of hospitalization for elective vascular surgery on the incidence rates of hospital admissions forcomplications caused by common vascular conditions compared to the pre-pandemic period.”.

“…the frequency of admissions for…” or “…hospitalization for…” was added in the Results and Discussion sections, where needed.

Early identification of life-threatening vascular conditions, which are mostly asymptomatic, is key, to be revised to is critical or essential. 

R: Early identification of life-threatening vascular conditions, which are mostly asymptomatic, is key essential.

Reviewer 2 Report

This is well written paper describing impact of Covid 19 on vascular surgery activity in south Italy. This paper can be interesting particularly in comparison with other geografical zone . To have notice of the outcome of the patients treated would increase the interest.

Author Response

This is well written paper describing impact of Covid 19 on vascular surgery activity in south Italy. This paper can be interesting particularly in comparison with other geografical zone . To have notice of the outcome of the patients treated would increase the interest.

R: We thank the reviewer for this suggestion. Of course, the multicenter follow-up of thousands of treated patients would take a long time, carrying the risk that this study would become obsolete. However, the long term follow-up of these patients will be the purpose of our next study.